# MNase Digestion Protection Patterns of the Linker DNA in Chromatosomes

**DOI:** 10.3390/cells10092239

**Published:** 2021-08-29

**Authors:** Chang-Hui Shen, James Allan

**Affiliations:** 1Biology Department, College of Staten Island, City University of New York, 2800 Victory Boulevard, Staten Island, NY 10314, USA; 2Biochemistry and Biology Ph.D. Program, Graduate Center, City University of New York, New York, NY 10016, USA; 3Institute for Macromolecular Assemblies, City University of New York, New York, NY 10031, USA; 4MRC Human Genetics Unit, Institute of Genetics and Cancer, University of Edinburgh, Western General Hospital, Crewe Road, Edinburgh EH4 2XU, UK; jallan@ed.ac.uk

**Keywords:** linker histones, linker histone globular domains, chromatosome, monomer extension, nucleosome position, chromatosome position, MNase digestion

## Abstract

The compact nucleosomal structure limits DNA accessibility and regulates DNA-dependent cellular activities. Linker histones bind to nucleosomes and compact nucleosomal arrays into a higher-order chromatin structure. Recent developments in high throughput technologies and structural computational studies provide nucleosome positioning at a high resolution and contribute to the information of linker histone location within a chromatosome. However, the precise linker histone location within the chromatin fibre remains unclear. Using monomer extension, we mapped core particle and chromatosomal positions over a core histone-reconstituted, 1.5 kb stretch of DNA from the chicken adult β-globin gene, after titration with linker histones and linker histone globular domains. Our results show that, although linker histone globular domains and linker histones display a wide variation in their binding affinity for different positioned nucleosomes, they do not alter nucleosome positions or generate new nucleosome positions. Furthermore, the extra ~20 bp of DNA protected in a chromatosome is usually symmetrically distributed at each end of the core particle, suggesting linker histones or linker histone globular domains are located close to the nucleosomal dyad axis.

## 1. Introduction

In the eukaryotic genome, DNA is packaged inside nuclei by association with histone proteins to form chromatin. The fundamental unit of chromatin is the nucleosome core particle which is comprised of two copies of each of the core histones H2A, H2B, H3 and H4 and ~147 base pairs (bp) of DNA [1]. Nucleosomes play a critical role in gene regulation through their ability to prevent the access of *trans*-acting factors to DNA (for review, see [2,3,4,5,6]). Gene activation and transcription are often accompanied by the removal of nucleosomes from the transcription factor binding region. As such, nucleosome positioning with respect to the underlying DNA is crucial for chromatin dynamics during gene activation.

In addition to the core histones, present as an octameric unit, linker histones are invariably associated with nucleosomes. These histones bind to the nucleosome, interacting with an extra 20 bp of linker DNA, to form chromatosomes [7,8,9]. Chromatosomal arrays are hierarchically folded into higher-order structures which can modulate nucleosome positioning [10]. Linker histones can be considered as epigenetic regulators as they bind reversibly to nucleosomes and alter chromatin structures and dynamics. For example, the removal of linker histones from chromatin can lead to the migration of the histone octamer and alterations in global nucleosome spacing and local chromatin compaction [11,12,13,14]. Consequently, linker histones are thought to be capable of influencing nucleosome positioning by the virtue of their role as agents of higher-order folding [8,9,15,16].

H1 linker histones are developmentally regulated proteins and, as a family, display a much higher sequence variability between species than the highly conserved core histones [17,18]. All linker histone H1 variants consist of three parts: a short disordered amino-terminal domain, a central highly conserved globular domain with a rigid structure, and a long highly disorganized carboxyl-terminal domain of high, but variable, lysine and arginine content. The linker histone-induced folding of chromatin into a higher-order structure is largely a function of the basic C-terminal domain of the protein once correctly located by the binding of the globular domain close to the dyad of the nucleosome [16]. The reversible binding of linker H1 to a nucleosome is directly related to many biological processes [19,20]. Furthermore, linker histone H1 variants are modified and regulated by various post-translational modifications, which in turn are thought to modulate the chromatin structure [21]. Several studies have also shown that linker histones contribute to the maintenance of chromatin compaction and life span in yeast [22,23,24].

The association between linker histones and nucleosomes is very dynamic [25,26,27], and the nature of this interaction could allow the linker histones or the core particle octamer to be replaced or removed and, thus, impact many biological and physiological functions. Many attempts have been attempted to determine the exact position of linker histones or the linker histone globular domains within the chromatosome and its precise interaction with the linker DNA. In general, two major classes of binding mode based on the location of the linker histone within a chromatosome were observed in early studies. In the on-dyad location, the linker histone globular domain binds to the nucleosomal DNA at the dyad and interacts with both linker DNAs equally [9,28,29,30]. In the off-dyad configuration, the linker histone globular domain binds in a DNA groove located off the dyad axis and protects the two linker DNAs unequally [31,32,33,34,35]. Recent structural studies on isolated nucleosomes using linker histone variants from fruit fly, frog, chicken, mouse and human have produced various structural models for linker histones binding. For example, *Drosophila* H1 and human H1.4 bind to nucleosome near dyad in an asymmetric manner [36,37], while chicken linker histone H5, *Xenopus laevis* H1.0b and human H1.5 bind symmetrically at the centre of the dyad [38,39,40]. These diverse binding modes may reflect distinct properties of the linker histone variants or subtypes bound to the nucleosome, although it is probable that the interaction could also be influenced by the nature of the DNA-histone octamer substrate.

Despite the wealth of experimental and computational studies, several issues concerning linker histones and their binding in chromatin remain unclear. For example, can the interaction between the linker histone and a nucleosome directly influence its position without help from remodelling complexes? How different is the mode of binding of the linker histone in a polynucleosomal array compared to a single, isolated nucleosome? Finally, how do linker histones direct the higher-order packaging of chromatin and how are these proteins organised within the folded fibres?

To further advance our understanding of the linker histone–nucleosome interaction, we reconstituted core histone octamers onto a 4.4 kb region of the chicken adult β-globin gene, in the presence or absence of linker histones or linker histone globular domains. We then employed monomer extension to map the sequence-directed nucleosome positioning over extensive stretches of DNA. This procedure not only provides a high-resolution output, but also has the distinct advantage of involving nothing more complex than a simple, direct visual comparison of bands on high-resolution sequencing gels [41,42,43,44,45]. Our results show that the presence of linker histones or linker histone globular domains has no direct effect on nucleosome positioning. Furthermore, the extra 20 bp of DNA protected by linker histones or their globular domains in chromatosomes is usually, but not always, symmetrically distributed with respect to the core particle, suggesting that linker histone globular domains locate close to the dyad axis of a chromatosome.

## 2. Materials and Methods

### 2.1. Phagemid Construction and Preparation of Single Strand DNA

Appropriate DNA fragments of the chicken β-globin gene promoter region were cloned from plasmid pCARB 4.4 [46] into the polylinker of pBluescript phagemid vectors (Stratagene). All eight overlapping fragments were cloned into pBluescript KS(−) in both orientations by using one or two blunt cutting sites to generate the following phagemids for monomer extension: EcXmV (−1052 to +426; relative to the cap site of the β-globin gene; *Eco*RI-*Xmn*I), XmEcS (−1052 to +426; *Eco*RI-*Xmn*I), SmEc (−1052 to −110; *Eco*RI-*Sma*I), EcSm (−1052 to −110; *Eco*RI-*Sma*I), LA (−406 to +200; *Pvu*II-*Pvu*II), LE (−406 to +200; *Pvu*II-*Pvu*II), Max (−110 to +426; *Sma*I-*Xmn*I) and Xma (−110 to +426; *Sma*I-*Xmn*I) (Appendix A). All phagemids were transformed into *E.coli* strain DH11S for preparation of phagemid single-stranded DNA. *E.coli* DH11S (F+) cells carrying the recombinant pBluescript-globin plasmids were infected with M13KO7 helper phage to prepare the single-stranded DNA as described in the Supplementary Material [47].

### 2.2. In Vitro Nucleosome Reconstitution and Titration of GH1/GH5 to Reconstituted Chromatin

Core particle DNA was prepared from pCBA4.4 which comprises a 4.4-kb *Eco*RI-*Bam*HI fragment of chicken β-globin sequence (-1052 to +3369, relative to the transcription start site of the adult gene) in pBluescript KS [41,42]. The pCBA4.4 plasmid was reconstituted with core histone at a core histone/DNA ratio of 0.5:1 (wt/wt). The reconstitution mixture, which contained linear pCBA4.4 DNA, chicken erythrocyte core histones, 2 M NaCl, 1 × TE and 0.1 mM PMSF, was incubated at room temperature for 5 to 10 min and then dialysed against a linear 2 M to 0.4 M NaCl gradient in 1 × TE at 4 °C for 3 to 4 h. The reconstitution was then carried out by dialysis against a linear 0.4 M to 80 mM NaCl gradient (in 1 × TE with 0.1 mM PMSF) overnight at 4 °C. Aliquots of this reconstituted chromatin were then titrated with one molecule of recombinant GH1/GH5 or chicken H1/H5 per core histone octamer on ice for 60 min. The preparation of chicken erythrocyte core histones, linker histones H1/H5 and recombinant GH1/GH5 is described in the Supplementary Material. As a control, some of the chromatin was treated exactly the same as the GH1/GH5 reconstitute but no GH1/GH5 was added. After this, MNase was added to the mixture to 5 unit/mL and CaCl_2_ to 1 mM and then the digestion was proceeded by incubating the mixture on ice for 30 min followed by 105 s at 37 °C for trimming of the digestion products. Finally, EDTA was added to 15 mM to stop the digestion reaction. Chromatosomal and core particle DNAs were isolated from either a 7% polyacrylamide gel or a 4.5% agarose gel routinely.

### 2.3. In Vitro Nucleosome Reconstitution at 37 °C

Reconstitution was carried out at a core histone/DNA ratio of 0.5:1 (*w*/*w*). The reconstitution mixture, which again contained linear pCBA4.4 DNA, chicken erythrocyte core histones, 2 M NaCl, 1 × TE and 0.1 mM PMSF, was incubated at room temperature for 5 to 10 min and then dialysed against a linear 2 M to 0.5 M NaCl gradient in 1 × TE at 4 °C for 3 to 4 h. Subsequently, the reconstitute apparatus was moved to 37 °C, and GH1 or GH5 was added at the ratio of 1:1 to the core histone concentration. The reconstitution was continued by dialysis against a linear 0.5 M to 80 mM NaCl gradient (in 1 × TE with 0.1 mM PMSF) for 3 to 4 h at 37 °C. Finally, the reconstitution was further dialysed against 80 mM NaCl (in 1 × TE with 0.1 mM PMSF) for another 1 h at 37 °C. The samples were collected and stored at 4 °C until use.

### 2.4. Monomer Extension

Purified chromatosomal and core particle DNA molecules were 5′ end-labelled and used as heterogeneous populations of primers for extension on each of a set of 8 single-stranded phagemids containing overlapping sections of the 1.5-kb globin region. Monomer extension was performed as described previously [41,42]. Briefly, the 5′ end-labelled, alkaline-denatured pCBA 4.4 core particle or chromatosomal DNA was annealed to excess phagemid single-stranded DNA and extended by Klenow DNA polymerase in the absence or presence of appropriate restriction enzymes. For EcSm, LE, Xma, SmEc, LA and Max mapping constructs, the restriction enzyme *Xba*I was used. For XmEcS and EcXmV mapping constructs, restriction enzymes *Xba*I, *Sse8387* I and *Bsp*EI were used. Products formed during extension reactions were analysed by electrophoresis in 6% denaturing polyacrylamide gels.

### 2.5. Analysis of the Nucleosome Mapping

The gel analysis of the monomer extension experiments were analysed as in the previous study [42]. Briefly, quantitative densitometer scans were obtained for each extension reaction after PhosphorImager (Molecular Dynamics, Artisan Technology, Champaign, IL, USA) analysis of the dried gels: band sizes were determined after densitometry of phosphor images by reference to markers. The equation which converted from coordinates of markers to DNA size was determined by a six or higher-order polynomial. The correlation coefficient was greater than 99.9% (Appendix A). The correlation between DNA size and the coordinates of markers was determined for each gel independently. The lengths of extension products were converted by the application of the equation after the densitometry of phosphor images. The locations of the positioning site boundaries were determined by the lengths of extension products with respect to a unique restriction enzyme cutting site in the β-globin gene. Densitometry traces carried out in the absence of a restriction enzyme were considered as background for quantitation adjustment. Normalisation was determined by common nucleosome positioning sites within overlap sequence. The lengths of extra DNA associated with chromatosomes compared to core particles were obtained from the difference between the core particle positioning sites and corresponding chromatosome positioning sites.

## 3. Results

### 3.1. DNA Sequence-Directed Nucleosome Positioning Is Determined by the Core Histone Octamer and Is Not Altered by the Addition of Linker Histone Globular Domains

Recent developments in high throughput genome-wide nucleosome positioning technologies, such as MNase-seq, ChIP-seq and DNase-seq, can determine nucleosome positioning at the single-base pair resolution [48,49,50,51,52]. Furthermore, structural studies using nuclear magnetic resonance (NMR) spectroscopy and cryo-electron microscopy (cryo-EM), revealed the static contact between linker histones and nucleosomes [36,37,38,39,40,53]. However, the arrangement and location of linker histones or linker histone globular domains within long chromatin fibres has not yet been fully established.

Monomer extension is an approach that can be used to compare the translational positions adopted by core histone octamers and chromatosomes and, thus, to determine the extra linker DNA protected by the linker histones or linker histone globular domains. To map both the upstream and downstream boundaries of these nucleosomes, two sets of single-stranded mapping clones containing the region of the DNA to be analysed, in both orientations, were prepared. For our current study, we analysed nucleosome positioning on a 1.5 kb region of the chicken β-globin gene (Appendix A).

For the preparation of core particle DNA and chromatosomal DNA, a plasmid (pCBA4.4) containing the chicken adult β-globin gene with its enhancer and flanking sequences (4.4 kb) was reconstituted at a core histone/DNA ratio of 0.5:1 (*w*/*w*) [42]. Under our reconstitution conditions, the amount of core histones added to the DNA was far from saturating. This is because we could identify the strongest core histone octamer binding sites on the template. Furthermore, we could avoid both histone–histone interactions between core particles and cooperativity in histone octamer binding to DNA. If we used a higher histone/DNA ratio, these activities would influence nucleosome placement in reconstitutes and it would be difficult to identify where the core histone octamer binding sites were. To generate chromatosomes, core histone reconstituted chromatin was titrated with one molecule of recombinant linker histone globular domain (GH1 or GH5) per core histone octamer (Appendix A). Control chromatin was treated in the same manner as the globular domain reconstitutes but no GH1/GH5 was added. Reconstituted chromatins were then digested with MNase to produce populations of chromatosome and core particle DNAs (Figure 1A). One clear discrete band with size of ~147 bp was observed in the absence of linker histone globular domains indicating the presence of core particle DNA (Figure 1A lane 2). In the presence of either GH1 or GH5, two bands were observed: one band with a size of ~147 bp was core particle DNA and the other with a size of ~167 bp was chromatosomal DNA (Figure 1A lanes 3–4)). For the 147 bp DNA band obtained from GH1- or GH5-containing reconstitutes, they might have come from two different populations: one set of the 147 bp DNAs might have been from chromatosomes that were over digested by MNase and had lost GH1 or GH5; the other set of 147 bp DNA might have come from core particles that have never been bound by GH1 or GH5. These DNAs were purified and analysed in a 6% denaturing polyacrylamide gel (Figure 1B).

Purified nucleosomal DNAs (“monomer DNAs”) were then employed as primers in the monomer extension procedure so as to determine their corresponding core particle or chromatosome positions at high resolution (Figure 1C). Briefly, a plasmid carrying the sequence (β-Globin gene, for example) to be mapped was reconstituted with core histones and then digested with micrococcal nuclease to generate mononucleosomes (core particle). Monomer DNA, purified from these mononucleosomes, agarose gel-extracted, was 5′ end-labelled with a T4 polynucleotide kinase, which generated the 3′ OH essential for priming. This 5′ end-labelled monomer DNA was then annealed to a single-stranded (ss) DNA template (the EcXmV, for example) prepared from a phagemid carrying the sequence to be mapped and extended by the addition of DNA polymerase and dNTPs. This reaction was carried out in the presence of a restriction enzyme, which cleaved, at a unique site (*Xba*I, for example), the double-stranded DNA formed upon the extension of the monomers. As a result, labelled fragments were produced, the length of which located the 5′ end of the annealed monomer DNAs relative to the known location of the restriction site employed; thus, enabling one nucleosome boundary to be defined. The other boundary was mapped by annealing and extending the same monomer DNA on the complementary ssDNA template (the XmEcs, for example), using an appropriate restriction site (*Xba*I, for example) as the reference point.

An example gel analysis of the monomer extension products obtained by mapping both the upstream and downstream boundaries of nucleosomes formed in reconstituted chromatin, is shown in Figure 2A,B. In the presence of restriction enzyme *Xba*I, the extension of the monomer DNA gave rise to a set of discrete, monodisperse bands (Figure 2A, lanes 10–14; Figure 2B, lanes 3–7). Reactions performed in the absence of the restriction enzyme gave rise to extension products comprising only high molecular weight DNAs and no discrete bands could be detected (Figure 2A, lanes 3–7). The discrete extension products produced in the presence of *Xba*I were, therefore, indicative of the boundaries of nucleosome positioning sites. Our monomer extension results showed clearly that the chromatosome positioning pattern (Figure 2A, lanes 12 and 14; Figure 2B, lanes 5 and 7) was different from the core particle positioning pattern (Figure 2A, lanes 10, 11 and 13; Figure 2B, lanes 3, 4 and 6). Most of the bands in the core particle positioning site lanes were accompanied by bands in the chromatosome positioning site lanes, which were correspondingly larger. There appeared to be no independent chromatosome sites lacking corresponding core particle sites. Thus, the addition of GH1/GH5 did not generate new positioning sites.

We also noted that the core particle positioning sites derived from reconstitutes lacking linker histone globular domains (Figure 2A, lanes 10; Figure 2B, lanes 3) and the core particle positioning sites derived from reconstitutes which contained GH1 or GH5 (Figure 2A, lanes 11 and 13, respectively; Figure 2B, lanes 4 and lanes 6, respectively) were generally similar in size. Although small differences in the relative abundance (band intensity) of particular sites derived from different chromatins could be found, the location of corresponding core particle positioning sites was constant. Therefore, core particles derived from chromatosomes occupied the same positions on the DNA as core particles formed in the absence of globular domains.

We also observed that chromatosome positioning sites derived from a GH1-containing reconstitute (Figure 2A, lane 12; Figure 2B, lane 5) were very similar to the chromatosome positioning sites derived from a GH5-containing reconstitute (Figure 2A, lane 14; Figure 2B, lane 7). Generally speaking, GH1 and GH5 chromatosome bands seemed to be the same under our experimental conditions, although in a few instances, these may have differed in relative abundance. Thus, the formation of chromatosome positioning sites seemed to be independent of two types of linker histone globular domains used in this study.

By combining all of the mapping analyses, translational positioning maps for the histone octamer and GH1- and GH5-containing chromatosomes, covering the entire 1.5 kb analysed, were generated (Figure 2C). The core particle positioning map was similar to that found in a previous study [42], although there were minor quantitative differences which may be attributed to a variation in mapping constructs or to a variation in core particle preparation.

### 3.2. Effect of Reconstitution Conditions on Nucleosome Positioning

Positioned nucleosomes have been shown to display a dynamic behaviour, interpreted to indicate movement between different positioning sites [54,55,56]. These observations, suggest that temperature-induced, DNA sequence-dependant nucleosome mobility is a general phenomenon [55,56]. Our results demonstrated that the addition of linker histone globular domains to reconstituted chromatin did not alter nucleosome positioning. Instead, the chromatosome positioning appeared to be determined by the core particle positions established before the addition of linker histone globular domains during the reconstitution process. As almost all of the chromatosome positions mapped can be attributed to established core positions, there is little evidence to suggest that new positions resulted from the binding of linker histone globular domains (Figure 2).

As the studies described above were carried out by the addition of GH1 or GH5 to the reconstitutes at 4 °C in 80 mM NaCl, conditions that may not be particularly favourable for octamer mobility, we repeated the reconstitution at an elevated temperature. Linker histone globular domains were added during the reconstitution process, when the salt concentration had been reduced to 0.5 M NaCl, and then the remaining dialysis to 80 mM NaCl was performed at 37 °C. Chromatin prepared in this manner was then digested with MNase under standard conditions to produce chromatosome and core particle DNAs (Appendix A).

Results from these experiments showed that both core particle and chromatosome positioning sites derived from chromatin reconstituted at 37 °C (Figure 2A, lanes 17–21; Figure 2B, lanes 8–12) were, in general, very similar to the positioning sites derived from chromatin reconstituted at 4 °C (Figure 2A, lanes 10–14; Figure 2B, lanes 3–7). Thus, core particle and chromatosome positioning was not influenced substantially by the temperature at which the final stages of reconstitution were carried out or by the salt concentration at which GH1/GH5 was added. It is also possible that our reconstitution process may not have provided enough dynamic conditions for the occurrence of nucleosome repositioning.

### 3.3. MNase Digestion Patterns of the Linker DNA in Chromatosomes

We analysed the lengths of GH1/GH5-induced DNA extensions in chromatosomes, relative to core particles, for the majority of positioning sites identified throughout the 1.5 kb region mapped. The length difference between a chromatosome band and the core particle band is a measure of the length of DNA extending from one side of the core particle and protected from digestion in a chromatosome by the linker histone globular domain. To characterise this feature in detail, DNA extensions derived from GH1 chromatosomes were carefully measured to determine the distribution of lengths.

For GH1 chromatosomes, the majority of DNA extension lengths exhibited a distribution centred at 10 bp with an average DNA length of 10.4 ± 0.40 bp (Figure 3A). This suggested a 10 + 10 bp extra DNA protection pattern in chromatosomes. However, a few chromatosomes displayed a more varied combination of paired extension lengths (Appendix A). By summing the paired lengths of DNA extensions for each of the same 38 nucleosomes, the total length of additional DNA associated with a chromatosome (compared to a core particle) averaged 20.8 ± 0.45 bp (Figure 3B). These results indicated that the DNA extended from a core particle to form chromatosome amounts to about 21 bp and were usually equally distributed in 10 bp units at both ends of the core particle. Thus, the extra DNA protected by linker histone globular domains was usually symmetrically distributed at each side of the core particle.

### 3.4. Effects of the Linker Histones Tails on the Nucleosome Positioning

Although the structure and function of the N- and C-terminal domains of linker histones remain poorly understood, in part because of their intrinsically disordered nature, a recent cryo-EM experimental study demonstrated a role for the C-terminal domain in stabilizing the H1-nucleosome complex by primarily binding to one of the linker DNA arms [40]. Furthermore, a more extensive study using cryo-EM, NMR and MD simulation to study three human H1 isoforms confirmed that the C-terminal domain of linker histone H1 facilitated a stabilizing effect on DNA dynamics and the binding dynamics between H1 and core histone tails [57]. To characterise the influence of the full-length linker histones upon nucleosome positioning and chromatosome formation, we compared (i) linker histone globular domains with intact linker histones and (ii) intact linker histone subtypes, with respect to these properties. Again, monomer extension was employed using reconstituted chromatin prepared with intact chicken erythrocyte H1 or H5.

To generate chromatosomes, core histone reconstituted chromatin was titrated with one molecule of recombinant linker histone (H1 or H5) per core histone octamer as described previously. Control chromatin was treated in the same manner as the linker histones reconstitutes but no H1/H5 was added. Reconstituted chromatins were then digested with MNase to produce populations of chromatosome and core particle DNAs (Appendix A). Purified nucleosomal DNAs (“monomer DNAs”) were then employed as primers in the monomer extension procedure. Our results showed that the H1- and H5-containing chromatosome positioning pattern (Figure 4, lanes 5, 7, 16 and 18) was very similar to the corresponding chromatosome positioning sites derived from GH1 or GH5-containing reconstitutes (Figure 4, lanes 2, 3, 13 and14). No novel chromatosome positioning sites without corresponding core particle sites were formed, indicating that the addition of H1/H5 did not generate new positioning sites. Although there were some minor quantitative differences between the corresponding sites, the patterns of positioning formed in the presence of globular domains or intact linker histones were very similar. In addition, the chromatosome positioning sites obtained from H1-reconstitutes were equivalent to chromatosome positioning sites derived from H5-reconstitutes. These results suggest that, under the reconstitution conditions employed here, the linker histone tails did not have a substantial influence upon nucleosome positioning and that the linker histone globular domains determined the nature of the chromatosomal DNA extensions.

## 4. Discussion

The binding of the globular domain of linker histones to nucleosomes is a critical step in the pathway by which these molecules interact with chromatin, and modulate, induce and maintain a higher-order chromatin structure [8,9,15,16,58]. The capacity of linker histone tails to fold nucleosomes into a higher-order structure depends upon correct globular domain binding [16]. Since the higher-order chromatin structure imposes a constraint on nucleosome positioning, linker histone could in principle at least indirectly affect positioning as it is the agent of higher-order folding.

In the present study, we examined the translational positioning maps for the histone octamer, and GH1- and GH5-containing chromatosomes, which formed on a 1.5 kb stretch of DNA after reconstitution (Figure 2). Our core particle positioning map was similar to that previously described [42]. We also demonstrated that the addition of linker histone globular domains or full-length linker histones to core histone reconstituted chromatin did not appear to alter nucleosome positioning (Figure 2 and Figure 4). Instead, chromatosome positioning sites appeared to be determined by the core particle sites established during the reconstitution process before the addition of linker histone globular domains or linker histones. Almost all the chromatosome positions which were mapped could be accounted for on the basis of the established core positions. There was little evidence to suggest that new positions were being determined by the presence of linker histone globular domains or linker histones.

Since GH1 and GH5 share the same structure, three helices followed by a β-hairpin and were similarly folded, it was perhaps not surprising that GH1 and GH5 should have displayed similar nucleosome binding properties (Appendix A). On the other hand, intact H1 and H5 molecules were arranged along the chromatin fibre in a polar, head-to-tail arrangement [59,60]. The contact between C- and N-terminal tails raised the possibility of a capacity to influence nucleosome positioning through bridging neighbouring nucleosomes. Furthermore, the distinctive arginine-rich C-terminal tail of H5 had a higher affinity for chromatin than the lysine-rich C-terminal tail of H1 which resulted in a more stable H5-bound chromatin structure than H1-bound chromatin structure [26]. It has been shown that the H3 N-terminal tail interacts with flanking DNA sequences in the absence of H1 whereas, in the presence of H1, the linker histone competes with the H3 tail for binding to these flanking DNA sequence and forces the H3 tail to interact with DNA in the nucleosomal core [61]. Additionally, the binding of linker histones to the nucleosome can disrupt the symmetric conformation of the H2A C-terminal tails in the same nucleosome, probably due to the stabilisation of the flanking DNA sequences by H1 [57]. Therefore, one might expect H1 and H5 to display differing effects on nucleosome positioning. However, we observed that chromatosome positioning sites were very similar between H1-containg reconstitutes and H5-containing reconstitutes, suggesting that both H1 and H5 had no effects on nucleosome positioning. Nevertheless, we did observe small changes in the binding affinity in some nucleosomes derived from linker histone-containing reconstitutes when compared to those derived from reconstitutes without linker histones. It is possible that the regions where linker histones showed a weak affinity could have biological consequences for DNA-based cellular activities such as replication, transcription, recombination and repair.

Under our reconstitution conditions, the amount of core histones added to the DNA was far from saturating and should have resulted in the formation of about one core particle per 500 bp of DNA. Although the level of linker histone globular domain or intact linker histone added during reconstitution equated to one molecule per core particle, competition by the excess DNA may effectively reduce this one-to-one ratio, so that some nucleosomes might not have linker histones or linker histone globular domain bound. Thus, the opportunity for linker histone–linker histone, or linker histone tail interactions are likely to be very limited. Nevertheless, our results did show that, in relative isolation, individual linker histones did not have the capacity to alter established core particle positions.

It is generally accepted that linker histones bind to the nucleosome and protect an extra 20 bp of linker DNA. Many explanations have been proposed to describe the location of the linker histones, or linker histone globular domains, within the nucleosome, their interaction and chromatosome protection through biochemical analysis, structural studies or computing simulations [9,36,37,38,39,40,57,62,63,64,65,66,67]. The current study is the first in vitro analysis of chromatosomal DNA extensions protected by linker histones or linker histone globular domains, in the context of a long-range nucleosome array formed on a natural genomic DNA sequence. Although the monomer extension data was not quantitively as accurate as a high throughput sequencing analysis might provide, our study allowed a direct visual comparison on the impact of linker histones or linker histone globular domains on nucleosome positioning and DNA extensions. An analysis of chromatosomal DNA extensions from a large population of chromatosomes (38 sites; Supplementary Material) showed that most of the sites adopted 10 + 10 bp extensions with respect to the protection from core particles (Figure 3). This arrangement was entirely consistent with GH5 being placed between one terminus of chromatosomal DNA and the DNA in the vicinity of the dyad axis of symmetry of the core particle [31]. In this arrangement, the linker histone globular domains would be able to bind the nucleosome-linking DNA strands that exit and enter the nucleosome simultaneously and, therefore, protect a symmetrical 10 bp extension of DNA. Our observation also suggested that linker histones or linker histone globular domains generally protected DNA exiting from the core particle in an apparently symmetrical manner with relatively few exceptions. However, it has been shown that, in some instances, histone octamer binding alone can transiently protect additional DNA at the upstream and/or downstream ends of core particle binding sites [49]. This property may influence the binding of linker histones or linker histone globular domains and contribute to the nature of the extra DNA protected in the chromatosome. Additionally, it is likely that bending energy differences between the two protected regions at the edges of positioning sites may modulate the symmetry of protection.

Recently, Bednar et al. (2017) performed a hydroxyl-radical footprinting analysis and showed that both full-length H1 (H1.0) and the isolated GH1 (GH1.5) domains make a symmetric footprint on the core DNA, protecting the central base pair plus three to four flanking nucleotides on each strand [40]. Their results confirmed that the linker histone H1.0 and linker histone globular domain GH1.5 adopts an on-dyed binding mode in the solution. These findings are in line with the footprinting pattern observed for the binding of H1.5 to di- and tri-nucleosomes in their early study [28]. As such, the observed protection pattern in the solution agrees well with the specific protein-DNA interfaces in their crystal structure and with the effect of H1 on linker conformation seen by their cryo-EM result [40]. This on-dyed binding configuration is also observed for the globular domains of chicken H5 [38]. Although we did not perform site-specific cross-linking and DNA footprinting experiments to examine specific interactions between nucleosome and either linker histones or linker histone globular domains, our MNase digestion protection and monomer extension analysis indicated that linker histones and linker histone globular domains usually protected linker DNA symmetrically.

All the on-dyad binding configurations observed for the chicken GH5 [38], *Xenopus* H1.0 [40], human GH1.5 [40], chicken H1/H5 (this study) and chicken GH1/GH5 (this study), differed from the off-dyad binding reported for the human H1.4 globular domain in condensed 12-nucleosome arrays [37]. Several reasons can explain the discrepancy. First, H1.4 and H1.5 have 95% sequence identity in their globular domain, and the few divergent residues are unlikely to account for the different binding configurations [40]. It is likely that it may be due to the variations in the process of sample preparation, such as the use of different cross-linking reagents in these two different studies or the use of different reaction conditions. Secondly, the 12-nucleosome arrays showed a different H1 binding mode from our H1/H5 observation which was also a long-range nucleosome array. This might have come from the different methodologies used in the observation. The cryogenic electron microscopy may not be equivalent to nuclease digestion which measures a limiting structure. Furthermore, the chromatin used in their study is a tetra-nucleosome which is capable of forming a higher-order chromatin structure and, under these circumstances, the tails of H1 may have an inter-nucleosome effect and may influence the location of the globular domain in their study. On the other hand, our reconstitution condition was to avoid such an interaction. It is believed that the local stereochemical constraints play an important role in the adoption of different binding modes and linker histones in condensed array are easy to adopt asymmetrical binding. This is due to the twisted fibre geometry of the array, which requires the two linkers of each nucleosome to follow non-superimposable trajectories as they connect to the preceding and subsequent nucleosome. Under this circumstance, GH1.4 would adopt off-dyed binding to stabilize the nucleosome array structure and form a higher-order chromatin structure. Therefore, it is likely that linker histone binding mode can switch from on-dyed to off-dyed and vice versa during the chromatin condensation process [38,40].

In conclusion, we showed that nucleosome positioning is determined by the DNA sequence-directed binding of the core histones and that this is not notably influenced by the subsequent binding of linker histones or linker histone globular domains. Furthermore, the majority of core particle positioning sites displayed a symmetric mode of chromatosome protection in our study. These observations indicated that the linker histones or linker histone globular domains bind close to the dyad axis of the nucleosome in a symmetric configuration and protect 10 bp at each terminus. As such, we provided some insight into the chromatosomal DNA extensions that could be established in a long-range nucleosome array and this may help to further understand linker histones’ dynamic interaction within chromatin.

## Figures and Tables

**Figure 1 cells-10-02239-f001:**
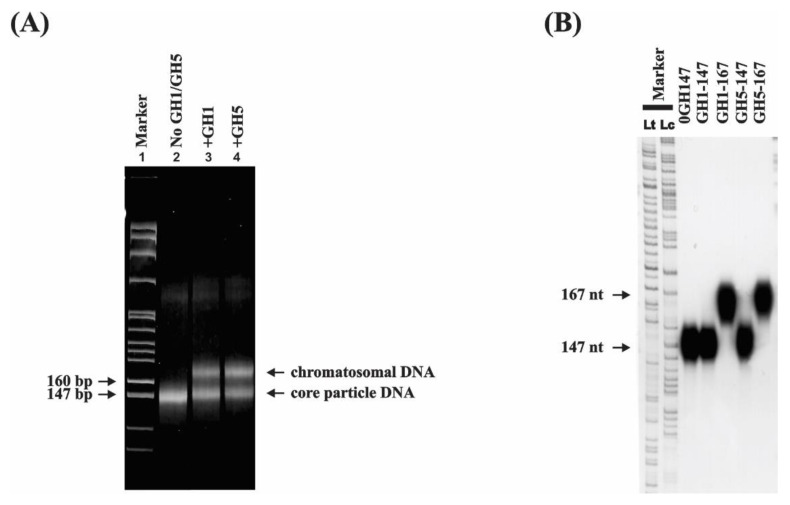
(**A**) Chromatosome protection of reconstituted chromatin containing no GH1/GH5 or equal molar ration of recombinant GH1 (+GH1) or GH5 (+GH5). DNA purified from MNase digested chromatin were run in a 6% polyacrylamide gel. The marker was an *Msp*I digest of pBR322 DNA. (**B**) A 6% denaturing polyacrylamide gel analysis of 5′ end labelled core particle and chromatosomal DNAs; 0GH147: core particle DNA derived from reconstitutes lacking linker histone globular domain. GH1-147 and GH5-147: core particle DNAs derived from GH1- and GH5-containing reconstitutes, respectively. GH1-167 and GH5-167: chromatosomal DNA derived from GH1- and GH5-containing reconstitutes, respectively. Markers: size standards included C and T sequencing reactions of M13mp18 DNA (Lc and Lt, respectively). (**C**) A schematic outline of the monomer extension and restriction procedures used to map nucleosome binding sites.

**Figure 2 cells-10-02239-f002:**
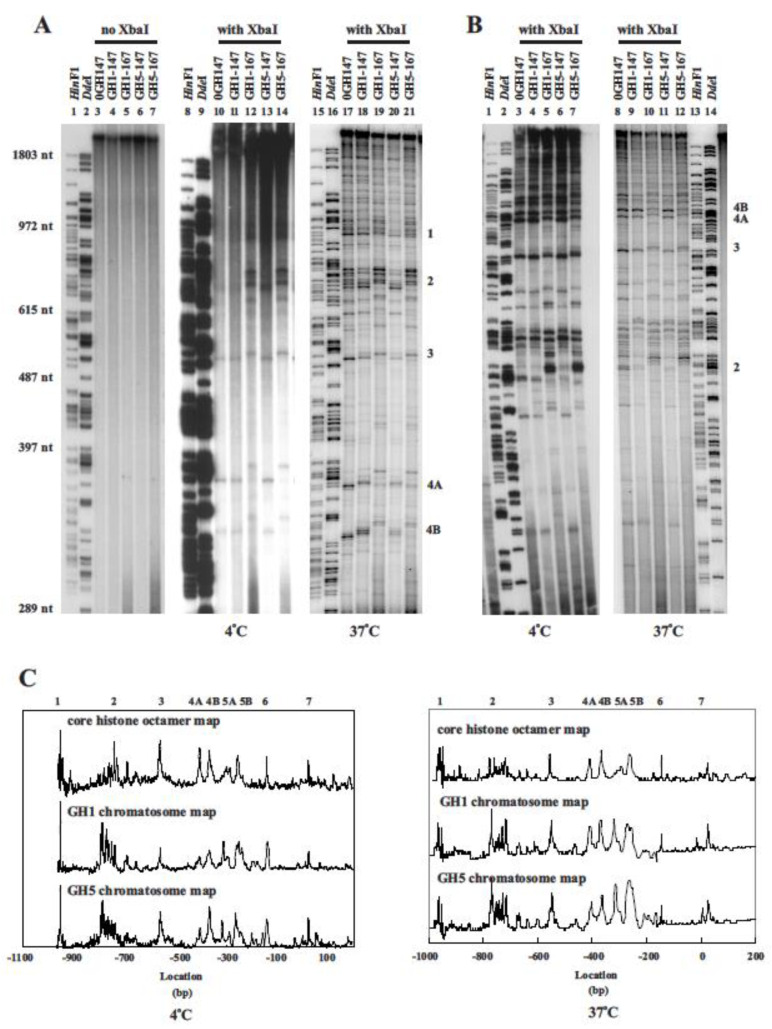
(**A**) A 6% denaturing polyacrylamide gel analysis of monomer extension products formed on mapping constructs EcSm (upstream) from 4 °C and 37 °C reconstitute condition. (**B**) A 6% denaturing polyacrylamide gel analysis of monomer extension products formed on mapping constructs SmEc (downstream) prepared from 4 °C and 37 °C reconstituted conditions. The names for each sample were as described in Figure 1. Size standards were *Hin*fI and *Dde*I digests of lambda DNA. Prominent core particle positioning sites or groups of sites previously identified were numbered as in Davey et al. (1995). (**C**) Maps of the histone octamer and chromatosome positioning sites for the chicken adult β-globin 5′ gene region (1.2 kb) prepared from 4 °C and 37 °C reconstituted conditions. The sequence was numbered with respect to the transcription start site of the gene. By assuming a core particle size of 147 bp and a chromatosome size of 167 bp, the maps were arranged to depict the centres of positioning sites. The names for samples from linker histone globular domain reconstitutes were as described in Figure 1.

**Figure 3 cells-10-02239-f003:**
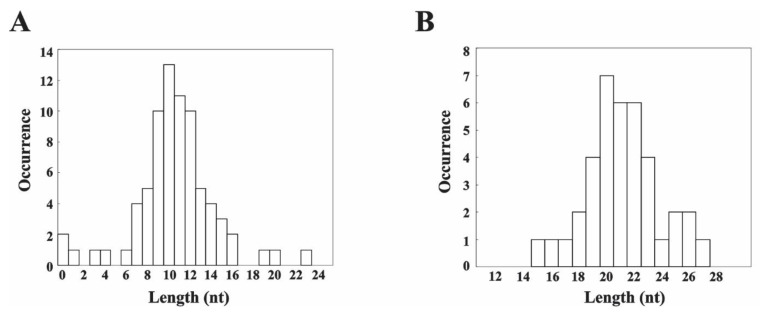
Distribution of lengths of extra DNA associated with chromatosomes compared to core particles. (**A**) Distribution for 76 different DNA extension lengths obtained from 38 selected nucleosomes. (**B**) Distribution for the sum of paired lengths of DNA extensions for each of the 38 nucleosomes. The coordinates of each of the 38 nucleosomes are listed in the Supplementary Material.

**Figure 4 cells-10-02239-f004:**
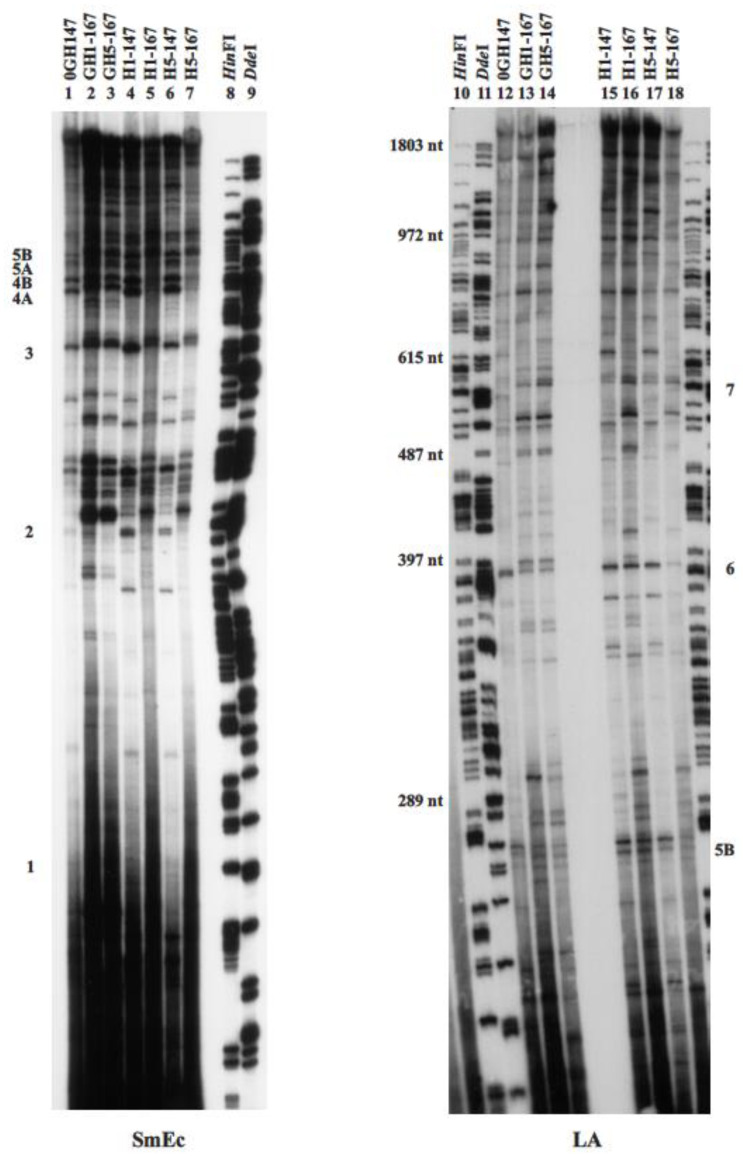
A 6% denaturing polyacrylamide gel analysis of monomer extension products formed on mapping constructs SmEc and LA from linker histone globular domains or linker histones reconstitutes. H1-147: core particle DNA derived from H1-containing reconstitutes. H1-167: chromatosomal DNA derived from H1-containing reconstitutes. H5-147: core particle DNA derived from H5-containing reconstitutes. H1-167: chromatosomal DNA derived from H5-containing reconstitutes. The names for samples from linker histone globular domain reconstitutes were as described in Figure 1. Size standards were as described in Figure 2. Prominent core particle positioning sites or groups of sites previously identified were numbered as in Davey et al. (1995) [42].

## Data Availability

Data is contained within the article or Appendix A.

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
