# Peer review of "MNase Digestion Protection Patterns of the Linker DNA in Chromatosomes"

_cells, 2021, doi:10.3390/cells10092239_

Round 1

Reviewer 1 Report

The authors map the sequence-directed nucleosome positioning over extensive stretches of DNA using monomer extension. They observe no direct effect of the presence of linker histones or linker histone globular domains on nucleosome positioning. They document that the extra 20 bp of DNA protected by linker histones or their globular domains in chromatosomes is usually symmetrically distributed with respect to the core particle and these linker histones are located close to the nucleosomal dyad axis.

This is a well written manuscript, using appropriate analytical methods to produce good data, and with appropriate interpretation of the data.

Minor:

Figure 1 is illegible. The size, resolution, and figure captions should be increased.

Author Response

The authors map the sequence-directed nucleosome positioning over extensive stretches of DNA using monomer extension. They observe no direct effect of the presence of linker histones or linker histone globular domains on nucleosome positioning. They document that the extra 20 bp of DNA protected by linker histones or their globular domains in chromatosomes is usually symmetrically distributed with respect to the core particle and these linker histones are located close to the nucleosomal dyad axis.

This is a well written manuscript, using appropriate analytical methods to produce good data, and with appropriate interpretation of the data.

Minor:

Figure 1 is illegible. The size, resolution, and figure captions should be increased.

We have replaced the original Figure 1 with the revised Figure with clear sketch and captions

Reviewer 2 Report

Summary:

In this study, the authors show that nucleosome positioning in a 1.5 kb region of the chicken β-globin gene is driven largely by the core histone octamer and that this positioning is unchanged in the presence of the globular and full-length chicken linker histones H1 and H5. In addition, the authors also determine that the H1 and H5 linker histones protect ~10bp of linker DNA on each end of the nucleosome, which is consistent with structural and biochemical assays previously performed by other labs. Importantly, establishing this with a sequence of in-vivo relevance in the absence of strong-positioning sequences is a highlight of this work. Overall, I feel this is a well-executed study that provides insight into the regulation of chromatin structure by linker histones and I think it will be well-received by the readership of “Cells”.

I do a few issues I feel the authors should address, and a couple comments I feel will make the paper more accessible to some readers:

Comments:

  1. The only major criticism I have of the paper is the rationale for some of the chosen experimental conditions, and potentially how the results may be altered if the experiments had been performed under slightly different conditions. I believe including additional data that has been collected to address the following, or providing a clear rationale for the conditions chosen would be of benefit to readers:

  • The chromatosome refolding was done at a 0.5 (wt/wt) ratio of core histone octamer to β-globin gene DNA. Did the authors perform a concentration gradient of core histone octamer to β-globin gene DNA? If not, I believe the rationale for using sub-stoichiometric amounts of histone octamer should be included in the main text of the paper (around paragraph 2, line 191). While I don’t believe these experiments are necessary, it would be useful to include in the supplement if they have already been performed to corroborate the findings.

  • Similarly, the monomer extension experiments were performed under a single concentration of chicken H1 or H5. Did the authors ever perform a titration of H1 or H5 concentrations? If so, I think this would be useful to include in the supplement as it would get at the question of whether the remaining 147 bp bands after MNase digestion are due to overdigestion and/or nucleosomes that do not contain H1 or H5 bound.

  • Did the authors ever perform the experiments at >100 mM NaCl? I’m wondering if a more dynamic positioning of the nucleosome+linker H1 or H5 would be observed under higher salt concentrations. I understand the experiments performed at 37oC likely provide some insight into this, and I do commend the authors for noting the limitations of the experimental conditions in that section of the text.

  1. The authors used chicken core histone octamers and linker histone H1 and H5. A supplemental figure showing sequence alignment of the chicken and human sequences of at least the linker histone H1 and H5 sequences would be of great use to readers to understand how this data may be interpreted in the context of human histones (I ended up doing this myself out of curiosity). I would also recommend delineating the globular vs. N-and C-terminal flanking regions.

  1. How were the chicken erythrocyte core histones obtained? This should be indicated in the methods.

  1. How were the recombinant H1 or H5 proteins obtained? This should also be included in the methods.

Author Response

In this study, the authors show that nucleosome positioning in a 1.5 kb region of the chicken β-globin gene is driven largely by the core histone octamer and that this positioning is unchanged in the presence of the globular and full-length chicken linker histones H1 and H5. In addition, the authors also determine that the H1 and H5 linker histones protect ~10bp of linker DNA on each end of the nucleosome, which is consistent with structural and biochemical assays previously performed by other labs. Importantly, establishing this with a sequence of in-vivo relevance in the absence of strong-positioning sequences is a highlight of this work. Overall, I feel this is a well-executed study that provides insight into the regulation of chromatin structure by linker histones and I think it will be well-received by the readership of “Cells”.

I do a few issues I feel the authors should address, and a couple comments I feel will make the paper more accessible to some readers:

Comments:

  1. The only major criticism I have of the paper is the rationale for some of the chosen experimental conditions, and potentially how the results may be altered if the experiments had been performed under slightly different conditions. I believe including additional data that has been collected to address the following, or providing a clear rationale for the conditions chosen would be of benefit to readers:

  • The chromatosome refolding was done at a 0.5 (wt/wt) ratio of core histone octamer to β-globin gene DNA. Did the authors perform a concentration gradient of core histone octamer to β-globin gene DNA? If not, I believe the rationale for using sub-stoichiometric amounts of histone octamer should be included in the main text of the paper (around paragraph 2, line 191). While I don’t believe these experiments are necessary, it would be useful to include in the supplement if they have already been performed to corroborate the findings.

 We did not do the reconstitution using a higher amount of core histone to DNA ratio such as 1:1. The reason is because our goal was to identify the highest binding sites on the template for core particle.  We also wanted to avoid the histone-histone interactions between core particles and cooperativity in histone octamer binding to DNA.  This is because these interactions would influence nucleosome placement in reconstitutes. This phenomenon would tend to supplant positioning directed solely by DNA sequence. Due to these reasons, we decided to employ 0.5 to 1 core histone to DNA ratio and we were able to identify the locations of the strongest binding sites throughout the gel analysis. If we used 1:1 core histone to DNA ratio during the salt gradient dialysis process, we may produce smear results and it would be difficult to identify where the core histone binding sites are.  This is because we will lose the ability to identify the discrete core particle binding sites.  In fact, this technique was developed previously to identify the strong nucleosome positioning sites in our lab.  (Ali Yenidunya, Colin Davey, David Clark, Gary Felsenfeld, James Allan (1994) Nucleosome Positioning on Chicken and Human Globin Gene Promoters in Vitro: Novel Mapping Techniques, JMB, 237, 401-414.).  In Yenidunya et al study, we did perform experiments with different core histone to DNA ratio and identify that the conditions with the ratio at 0.5 to 1 can limit and control the histone-histone interactions between core particles and cooperativity in histone octamer binding to DNA, and thus identified the strongest binding sites for core particles.  Our technique is developed to simply compare the influence of linker histones and linker histone globular domains to nucleosome positioning and chromatosomal protection. In this context, our approach has successfully accomplished the work. 

We have added the following statement to the main text. (Line:194)

Under our reconstitution conditions, the amount of core histones added to the DNA is far from saturating so that we were able to identify the strongest core histone octamer binding sites on the template.  Furthermore, we can avoid both histone-histone interactions between core particles and cooperativity in histone octamer binding to DNA.  If we used higher histone/DNA ratio, these activities would influence nucleosome placement in reconstitutes and it would be difficult to identify where the core histone octamer binding sites are.

  • Similarly, the monomer extension experiments were performed under a single concentration of chicken H1 or H5. Did the authors ever perform a titration of H1 or H5 concentrations? If so, I think this would be useful to include in the supplement as it would get at the question of whether the remaining 147 bp bands after MNase digestion are due to overdigestion and/or nucleosomes that do not contain H1 or H5 bound.

Although we did not perform a titration of H1 and H5 during the reconstitution, we did perform the titration with linker histone globular domains and we included the MNase digestion results in Supplementary data as Figure S3.  In the current study, we used 1:1 linker histones to core histone octamer ratio and 0.5:1 core histone octamer to DNA ratio.  Under this conditions, linker histone tail is not likely to reach to neighboring nucleosome and therefore linker histones would have no impact on the folding of the array of nucleosomes into higher order structure.  Furthermore, the amount of core histones added to the DNA is far from saturating and should result in the formation of about one core particle per 500 bp of DNA. Although the level of linker histone globular domain or intact linker histone added during reconstitution equates to one molecule per core particle, competition by the excess DNA may effectively reduce this one-to-one ratio, so that some nucleosomes might not have linker histones or linker histone globular domain bound.  Thus, the opportunity for linker histone-linker histone, or linker histone tail interactions are likely to be very limited.  Because of these reasons, we did have a statement as shown below about the source of 147-bp bands from linker histones/linker histone globular domains reconstitutes as shown below. (Lines: 208-215 )

“In the presence of either GH1 or GH5, two bands were observed: one band with size of ~147 bp is core particle DNA and the other with size of ~167 bp is chromatosomal DNA (Figure 1A lanes 3-4)).  For the 147bp DNA band obtained from GH1- or GH5- containing reconstitutes, they might come from two different populations:  one set of the 147 bp DNAs might be from chromatosomes that have been over digested by MNase and had lost GH1 or GH5; the other set of 147 bp DNA might come from core particles that have never been bound by GH1 or GH5. These DNAs were purified and analysed in a 6% denaturing polyacrylamide gel (Figure 1B).  “

  • Did the authors ever perform the experiments at >100 mM NaCl? I’m wondering if a more dynamic positioning of the nucleosome+linker H1 or H5 would be observed under higher salt concentrations. I understand the experiments performed at 37oC likely provide some insight into this, and I do commend the authors for noting the limitations of the experimental conditions in that section of the text.

            We did not perform experiment with high salt conditions and we added the following statement to the text. (Line: 320 )

It is also possible that our reconstitution process may not provide enough dynamic conditions for the occurrence of nucleosome repositioning.

  1. The authors used chicken core histone octamers and linker histone H1 and H5. A supplemental figure showing sequence alignment of the chicken and human sequences of at least the linker histone H1 and H5 sequences would be of great use to readers to understand how this data may be interpreted in the context of human histones (I ended up doing this myself out of curiosity). I would also recommend delineating the globular vs. N-and C-terminal flanking regions.

We have added the Figure S7 to illustrate the sequence alignment for chicken linker histones and human somatic linker histones.

  1. How were the chicken erythrocyte core histones obtained? This should be indicated in the methods.

Thank you for your reminder. We have added this part in the Supplementary Material.

The statement shown below is also added in Line 126.

“The preparation of chicken erythrocyte core histones, linker histones H1/H5, recombi-nant GH1/GH5 is described in the Supplementary Material.  “

  1. How were the recombinant H1 or H5 proteins obtained? This should also be included in the methods.

Thank you for your reminder. We have added this part in the Supplementary Material.

The statement shown below is also added in Line 126.

“The preparation of chicken erythrocyte core histones, linker histones H1/H5, recombi-nant GH1/GH5 is described in the Supplementary Material.  “